# Opinion: Optimizing climate models with process-knowledge, resolution, and AI

Tapio Schneider[1,2], L. Ruby Leung[3], and Robert C. J. Wills[4]

[1]California Institute of Technology, Pasadena, CA, USA
[2]Google Research, Mountain View, CA, USA
[3]Pacific Northwest National Laboratory, Richland, Washington, USA
[4]ETH Zurich, Zurich, Switzerland

**Correspondence:** Tapio Schneider (tapio@caltech.edu)

**Abstract.** Accelerated progress in climate modeling is urgently needed for proactive and effective climate change adaptation. The central challenge lies in accurately representing processes that are small in scale yet are climatically important, such as turbulence and cloud formation. These processes will not be explicitly resolvable for the foreseeable future, necessitating the use of parameterizations. We propose a balanced approach that leverages the strengths of traditional process-based parameterizations and contemporary AI-based methods to model subgrid-scale processes. This strategy employs AI to derive data-driven closure functions from both observational and simulated data, integrated within parameterizations that encode system knowledge and conservation laws. In addition, increasing resolution to resolve a larger fraction of small-scale processes can aid progress toward improved and interpretable climate predictions outside the observed climate distribution. However, currently feasible horizontal resolutions are limited to $O(10\,\mathrm{km})$, because higher resolutions would impede the creation of the ensembles that are needed for model calibration and uncertainty quantification, for sampling atmospheric and oceanic internal variability, and for broadly exploring and quantifying climate risks. By synergizing decades of scientific development with advanced AI techniques, our approach aims to significantly boost the accuracy, interpretability, and trustworthiness of climate predictions.

## 1 Introduction

Climate models serve two distinct purposes. First, they encode our collective knowledge about the climate system. They instantiate theories and provide a quantitative account of climate processes—the complex interplay of causes and effects that governs how the climate system operates. In this role, they belong to the realm of *episteme*, or explanatory science (Russo, 2000; Parry, 2021). Second, climate models function as practical tools that allow us to calculate how the climate system might behave under different circumstances that have not yet been directly observed. In this role, they fall under the realm of *techne*, or goal-oriented applied science (Russo, 2000; Parry, 2021). The requirements for climate models differ depending on their primary role as *episteme* or *techne*. As encodings of our understanding (*episteme*), climate models should strive for explainability and simplicity, even if it means sacrificing a certain level of accuracy. Understanding of the climate system at different levels of description emerges through a hierarchy of models, ranging from simpler ones such as one-dimensional radiative-convective equilibrium models to more complex ones such as atmospheric general circulation models with simplified

parameterizations of subgrid-scale processes (Held, 2005; Jeevanjee et al., 2017; Mansfield et al., 2023). On the other hand, as calculation tools (*techne*), climate models should aim to simulate the climate system as accurately as possible under unobserved circumstances.

Over the past six decades, climate modeling has operated under the tacit assumption that these two roles of climate models align, implying that the most complex models reflecting our understanding of the system are also the most accurate tools for predicting its behavior in unobserved conditions. This is a desirable goal, but it may not always be attainable in systems as complex as the climate system.

In this essay, we focus on climate models as *techne*, emphasizing their role as tools for accurately calculating the behavior of the climate system in unobserved circumstances, although, as we will see, this role cannot entirely be decoupled from *episteme*. The goal of calculating the behavior of the climate system is to obtain its statistics, including average temperatures at specific locations and seasons, the probability that daily precipitation in a given region exceeds some threshold, or the covariance between temperature and humidity, which can lead to potentially dangerous humid heat extremes. These calculations correspond to what Lorenz (1975) defined as predictions of the second kind, where future climate statistics are estimated given evolving boundary conditions, such as human-induced greenhouse gas emissions. This contrasts with predictions of the first kind, which focus on forecasting the future state of a system given its initial conditions $\zeta_0$, as seen in weather forecasting. Consequently, climate models as *techne* should aim to minimize a loss function of the form (Schneider et al., 2017a)

$$\mathcal{L} = \|\langle \boldsymbol{y}(t) \rangle - \langle \boldsymbol{\mathcal{H}} \circ \boldsymbol{\mathcal{G}}(t; \boldsymbol{\theta}, \boldsymbol{\lambda}, \boldsymbol{\nu}; \boldsymbol{\zeta}_0) \rangle\|_\Gamma^2. \tag{1}$$

Here, the angle brackets $\langle \cdot \rangle$ indicate an appropriate time averaging, such as a seasonal average over multiple years. The vector $\boldsymbol{y}(t)$ represents time-varying observables of the climate system, including those whose time average $\langle \boldsymbol{y}(t) \rangle$ gives rise to higher-order statistics such as the frequency of exceeding a daily precipitation threshold in a specific region. It may also include frequency-space observables, such as the amplitude and phase of the diurnal cycle of precipitation. The climate model, denoted as $\boldsymbol{\mathcal{G}}(t; \boldsymbol{\theta}, \boldsymbol{\lambda}, \boldsymbol{\nu}; \boldsymbol{\zeta}_0)$, is a mapping that takes parameter vectors $(\boldsymbol{\theta}, \boldsymbol{\lambda}, \boldsymbol{\nu})$ and an initial condition vector $\boldsymbol{\zeta}_0$ (usually important only for slowly varying components of the climate system, such as oceans and ice sheets) to time-varying simulated climate states $\boldsymbol{\zeta}(t) = \boldsymbol{\mathcal{G}}(t; \boldsymbol{\theta}, \boldsymbol{\lambda}, \boldsymbol{\nu}; \boldsymbol{\zeta}_0)$. The observation operator $\boldsymbol{\mathcal{H}}$ maps simulated climate states $\boldsymbol{\zeta}(t)$ to the desired observables $\boldsymbol{y}(t)$. Lastly, $\|\cdot\|_\Gamma = \|\boldsymbol{\Gamma}^{-1/2} \cdot \|_2$ represents a weighted Euclidean norm, or Mahalanobis distance. The weight is determined by the inverse of the covariance matrix $\boldsymbol{\Gamma}$, which reflects model and observational errors and noise due to fluctuations from internal variability in the finite-time average $\langle \cdot \rangle$. The weighted Euclidean norm is chosen because the climate statistics are aggregated over time, meaning that, due to the central limit theorem, it is reasonable to assume that these statistics exhibit Gaussian fluctuations (Iglesias et al., 2013; Schneider et al., 2017a; Dunbar et al., 2021). However, the specific choice of norm in the loss function is not crucial for the following discussion. The essence is that the loss function penalizes mismatches between simulated and observed climate *statistics*, with less noisy statistics receiving greater weight. This can be done for longer-term aggregate statistics or for shorter-term predictions, for example, of El Niño and its impact on the climate system. The relatively sparse statistics available from reconstructions of past climates can additionally serve as a useful test of climate models outside the distribution of the present climate (Zhu et al., 2022).

To achieve accurate simulations of climate statistics, the objective is to minimize the loss function (1) for unobserved climate statistics $\langle \boldsymbol{y} \rangle$ with respect to the parameters $(\boldsymbol{\theta}, \boldsymbol{\lambda}, \boldsymbol{\nu})$. Importantly, the climate statistics may fall outside the distribution of observed climate statistics, particularly in the context of global warming projections. Therefore, the ability of a model to generalize beyond the distribution of the observed data becomes essential. Merely minimizing the loss over observed climate statistics or even driving the loss to zero in an attempt to imitate observations and pass a "climate Turing test" (Palmer, 2016) is not sufficient. Instead, fundamental science and data science tools, such as cross-validation and Bayesian tools, need to be brought to bear to plausibly minimize the loss for unobserved statistics.

In the loss function, we distinguish three types of parameters:

1. The parameters $\boldsymbol{\theta}$ appear in process-based models of subgrid-scale processes, such as entrainment and detrainment rates in parameterizations of convection. These parameters are directly interpretable and theoretically measurable, although their practical measurement can be challenging.

2. The parameters $\boldsymbol{\lambda}$ represent the characteristics of the climate model's resolution, such as the horizontal and vertical resolution in atmosphere and ocean models.

3. The parameters $\boldsymbol{\nu}$ pertain to AI-based data-driven models that capture subgrid-scale processes or correct for structural model errors, either within process-based models of subgrid-scale processes or holistically for an entire climate model (Kennedy and O'Hagan, 2001; Levine and Stuart, 2022; Bretherton et al., 2022; Wu et al., 2024). These parameters are neither easily interpretable nor directly measurable but are learned from data.

This distinction among the parameters is useful as it reflects three different dimensions along which climate models can be optimized. First, optimization can be achieved by calibrating parameters and improving the structure of process-based models that represent subgrid-scale processes such as turbulence, convection, and clouds. These processes have long been identified as a dominant source of biases and uncertainties in climate simulations (Cess et al., 1989; Bony and Dufresne, 2005; Stephens, 2005; Vial et al., 2013; Schneider et al., 2017b; Zelinka et al., 2020). Second, optimization can be accomplished by increasing the resolution of the models, which reduces the need for parameterization (Bauer et al., 2021; Slingo et al., 2022). Finally, optimization can be pursued by integrating AI-based data-driven models. These models have the potential to replace (Gentine et al., 2018; O'Gorman and Dwyer, 2018; Yuval and O'Gorman, 2020; Yuval et al., 2021) or complement (Schneider et al., 2017a; Lopez-Gomez et al., 2022) process-based models for subgrid-scale processes. Additionally, they can serve as comprehensive error-corrections for climate models (Watt-Meyer et al., 2021; Bretherton et al., 2022; Wu et al., 2024).

In the past two decades, efforts to optimize climate models have often focused on individual dimensions in isolation. For example, Climate Process Teams, initiated under the U.S. Climate Variability and Predictability Program, have concentrated on enhancing process-based models by incorporating knowledge from observational and process-oriented studies into climate modeling (Subramanian et al., 2016). The resolution of atmosphere and ocean models has gradually increased, albeit at a pace slower than the advances in computer performance would have allowed (Schneider et al., 2017b). More recently, there have been calls to prioritize resolution increase, aiming to achieve kilometer-scale resolutions in the horizontal, with the expectation

that this would alleviate the need for subgrid-scale process parameterizations, such as those for deep convection, and substantially increase the reliability of climate predictions (Bauer et al., 2021; Slingo et al., 2022). Moreover, there is a rapidly growing interest to advance climate modeling by using AI tools, broadly understood to include tools such as Bayesian learning, deep learning, and generative AI (e.g., Schneider et al., 2017a; Reichstein et al., 2019; Chantry et al., 2021; Watson-Parris, 2021; Balaji et al., 2022; Irrgang et al., 2022; Schneider et al., 2023).

Beginning with a review of recent advances in the goodness-of-fit between climate simulations and observed records, here we will explore the potential benefits and challenges associated with optimizing each of the three dimensions mentioned earlier. Our analysis will highlight the importance of adopting a balanced approach that encompasses progress along each dimension, as this is likely to yield the most robust and accurate climate models and the most trustworthy and usable predictions.

## 2 Evolution of climate models

The climate statistics $\langle y \rangle$ used in the loss function (1) can vary depending on the specific application. For example, a national climate model may prioritize minimizing the loss within a particular country. However, there are several climate statistics that are generally considered important and should be included in any comprehensive loss function. Two such examples are the top-of-atmosphere (TOA) radiative energy fluxes and surface precipitation rates.

The inclusion of TOA radiative energy fluxes is crucial because accurately simulating these fluxes is a prerequisite for accurately simulating changes in any climate statistic. After all, radiative energy is the primary driver of the climate system. Changes in radiative energy fluxes caused by changes in greenhouse gas concentrations drive global climate change; climate models must accurately simulate changes in these energy fluxes and their effect on multiple climate system components, from oceans and land surfaces to clouds. As a consequence, errors in radiative energy fluxes affect many aspects of a simulated climate, from wind to precipitation distributions. The balance of TOA radiative energy fluxes must also be closed to machine precision. A closed energy balance is necessary to achieve a steady climate in unforced centennial to millennial integrations in which tiny imbalances of the energy budget otherwise accumulate over $10^7$ discrete timesteps, leading to large-scale climate drift. The conservation requirements for climate predictions—for what John von Neumann called the "infinite forecast" (Edwards, 2010)—are more stringent than those for the short-term integrations needed for weather forecasting. Similarly, precipitation rates are of significant importance as they are part of what closes the water balance and they directly impact human activities. Achieving accurate simulations of precipitation rates relies on accurately simulating numerous subgrid-scale processes within the climate system. Therefore, precipitation is an emergent property that serves as a holistic metric to assess the goodness-of-fit of a climate model.

Figure 1 assesses the evolution of climate models over the past two decades in simulating the observed climatology of TOA radiative energy fluxes and precipitation rates, setting aside temporarily that the loss minimization should occur for unobserved records. The figure displays the median root mean square (rms) error between model seasonal climatologies and observations, with all data conservatively remapped to a common 2.5° latitude-longitude grid using Climate Data Operators

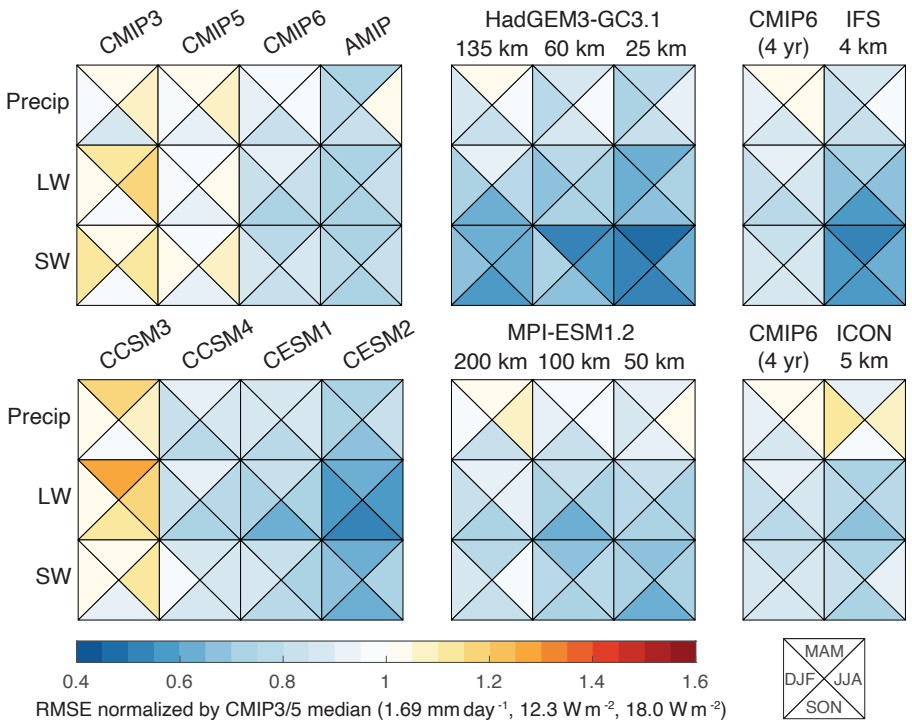

**Figure 1.** Normalized rms error (RMSE) in the seasonal climatology of precipitation, top-of-atmosphere (TOA) longwave (LW) radiation, and TOA shortwave (SW) radiation for different models and model intercomparison projects. The rms errors are relative to climatologies from GCPC (Adler et al., 2018) and CERES-EBAF (Loeb et al., 2009) datasets over the period 2001–2020. CMIP and AMIP rms errors represent median values of the RMSE computed separately for each of the included models. Climatologies are computed as follow: for CMIP3 over 2001–2020 for the B1 scenario; for CMIP5/6 over 2001–2020 for a combination of the historical and RCP4.5/SSP2.45 scenarios; for AMIP over 1995–2014 for models contributing to CMIP6; for HadGEM3-GC31 and MPI-ESM1-2 over 1995–2014 for the historical simulations; and over 4 simulated years for the kilometer-scale nextGEMS cycle 3 simulations (Koldunov et al., 2023), which are shown together with averages over the equivalent simulation length in CMIP6. The rms error is normalized by the median across CMIP3 and CMIP5 models for each field and across all seasons, with normalization constants shown below the colorbar. HadGEM3-GC3.1 and MPI-ESM1.2 from HighResMIP (Haarsma et al., 2016) are sorted in order of increasing horizontal resolution, with the atmospheric resolution for each configuration stated over the respective column (see Table 1).

(Schulzweida, 2023).[1] The plot includes three generations of climate models from the Coupled Model Intercomparison Project (CMIP) as well as recent higher-resolution simulations. It is evident that, over time, there has been a gradual improvement in
the fidelity of models in simulating TOA radiative energy fluxes and precipitation. For example, in CMIP6 (late-2010s), the median rms error relative to CMIP3 (mid-2000s) is reduced by 15% for precipitation, 31% for TOA outgoing longwave flux, and 30% for TOA reflected shortwave flux, with all values indicating average seasonal-mean improvements (Fig. 1, upper row).

---

[1]That is, what is displayed in Fig. 1 are unweighted errors, in contrast to the loss function (1), which downweights mismatches between simulations and observations for variables that have high error variance, e.g., because of internal variability in finite-time averages.

**Table 1.** Atmosphere and ocean model resolutions of HighResMIP simulations included in Fig. 1.

|  | Atmos. Res. | Ocean Res. | Vertical Levels |
|---|---|---|---|
| HadGEM3-GC3.1-LL | N96 (135 km) | 100 km | 85 |
| HadGEM3-GC3.1-MM | N216 (60 km) | 25 km | 85 |
| HadGEM3-GC3.1-HH | N512 (25 km) | 8 km | 85 |
| MPI-ESM1.2-LR | T63 (200 km) | 150 km | 47 |
| MPI-ESM1.2-HR | T127 (100 km) | 40 km | 95 |
| MPI-ESM1.2-XR | T255 (50 km) | 40 km | 95 |

Individual modeling centers have surpassed this median rate of improvement, for example, with rms error reductions of 30% for precipitation, 49% for TOA outgoing longwave flux, and 36% for TOA reflected shortwave flux in the progression from CCSM3 to CESM2 at the National Center for Atmospheric Research (NCAR) (Fig. 1, lower row). These improvements primarily stem from advances in process-based parameterizations and model tuning (e.g., Danabasoglu et al., 2020). The average resolution has also increased across the model generations, shifting from around 200–400 km horizontally in the atmosphere in CMIP3 to around 100–200 km in CMIP6 (Schneider et al., 2017b; Intergovernmental Panel on Climate Change, 2021).

To specifically examine the impact of resolution, we consider two models from the High Resolution Model Intercomparison Project (HighResMIP; Haarsma et al., 2016): HadGEM3-GC3.1 (Roberts et al., 2019) and MPI-ESM1.2 (Gutjahr et al., 2019). These models have conducted simulations at three different resolutions, with horizontal resolutions in the atmosphere between 25 and 200 km, without resolution-specific tuning (Table 1). Both models exhibit a modest but consistent reduction in error metrics as resolution increases. However, there is one exception: the doubling in atmospheric horizontal resolution from MPI-ESM1.2-HR (100 km) to MPI-ESM1.2-XR (50 km), without an increase in ocean resolution or atmospheric vertical resolution, did not result in an improvement in error metrics. This finding suggests that ocean resolution and atmospheric vertical resolution are also important factors contributing to the improvements with resolution.

Recently, there has been a push to increase the resolution of climate models even further to kilometer scales, allowing for partial resolution of deep convection and potential improvements in simulating precipitation and its extremes (Bauer et al., 2021; Slingo et al., 2022). In numerical weather prediction, enhanced horizontal resolution has led to improvements, for example, in rainfall predictions on timescales from hours to days (Clark et al., 2016). However, whereas assimilation of data at the initialization of a forecast continuously pulls numerical weather predictions close to the climate attractor, long-term climate simulations require a realistically closed energy balance to remain on the climate attractor. The energy balance also depends on dynamics at scales well below 1 km (e.g., in tropical low clouds, which are crucial for climate but less important for weather prediction), making it less clear that increased resolution by itself results in better climate simulations. Figure 1 displays the rms errors of two kilometer-scale coupled models (IFS and ICON) in simulating the seasonal climatology of TOA radiative

energy fluxes and precipitation.[2] For direct comparison to these short kilometer-scale simulations, Figure 1 includes the rms errors in CMIP6 simulations for the equivalent averaging period of 4 years. Compared to the coarser-resolution simulations, the kilometer-scale simulations show improvements in TOA shortwave fluxes and longwave fluxes, but little improvement or in some cases even increased errors in precipitation. These simulations highlight that higher resolution alone does not guarantee an improved fit in climate simulations. Many crucial climate-regulating processes, such as shallow clouds and cloud microphysics, remain unresolved at kilometer-scales, requiring appropriate parameterization. Extensive calibration or even re-design of subgrid-scale parameterizations at kilometer-scale resolution is necessary to reduce large-scale biases that can otherwise exceed those seen in coarser-resolution models (Wedi et al., 2020; Hohenegger et al., 2023). However, the high computational cost at kilometer-scale resolutions has so far inhibited systematic model calibration or exploration of novel parameterization approaches needed at these resolutions.

Figure 2 provides a more detailed illustration of how kilometer-scale models can inherit longstanding biases from coarse-resolution models. The figure compares August precipitation between observations and simulations. The simulations include coarse-resolution AMIP models and a set of kilometer-scale simulations conducted under the DYAMOND project (Stevens et al., 2019). The figure reveals that the kilometer-scale simulations capture more intricate details in the precipitation patterns, such as the strong orographic precipitation in the Himalayas, New Guinea, and the Sierra Madre Occidental. However, they still exhibit similar large-scale biases as the coarse-resolution simulations, such as excessive precipitation over the tropical regions of the south Pacific and Indian Oceans, commonly referred to as the double-ITCZ bias (Tian and Dong, 2020). The double-ITCZ bias has important implications for regional precipitation projections over land (Dong et al., 2021).

Over the past two decades, then, climate models have shown gradual improvements in key metrics, with error reductions of 10–20% per decade, as seen in Figure 1 and in other studies (Bock et al., 2020). However, there are still errors that are large compared to the climate change signals we aim to predict. For instance, the radiative forcing due to doubling $CO_2$ concentrations is about $4 \text{ W m}^{-2}$, while rms errors in TOA radiative energy fluxes are $O(10 \text{ W m}^{-2})$. The response of climate models to increasing greenhouse gas concentrations also varies widely across models. For example, the time when the $2°C$ warming threshold of the Paris agreement is exceeded varies by several decades among models (Schneider et al., 2017b; Intergovernmental Panel on Climate Change, 2021). This indicates that there is significant room for further improvement.

Given the significant errors in simulating the current climate and the uncertainties in future projections, there exists a large gap between the demands placed on climate models for adaptation decisions—such as designing stormwater management systems or sea walls to handle a 100-year flood in the decades ahead—and the capabilities of models today (Fiedler et al., 2021; President's Council of Advisors on Science and Technology, 2023). Yet, the need for such decision-making is immediate. Therefore, it is urgent to accelerate the improvement of climate models, aiming for a step change enhancement in both accuracy and usability for decision-making, beyond the gradual advances of recent decades. The question is how to achieve such a step change.

---

[2]NextGEMS cycle 3 output (Koldunov et al., 2023) was averaged over 2021-2024 in ICON to avoid the influence of SST nudging during the spin-up period and the subsequent ∼0.8°C drift in global-mean SST over the first year of simulation; the output was averaged over 2020-2022 and 2024 in IFS, due to missing data in 2023.

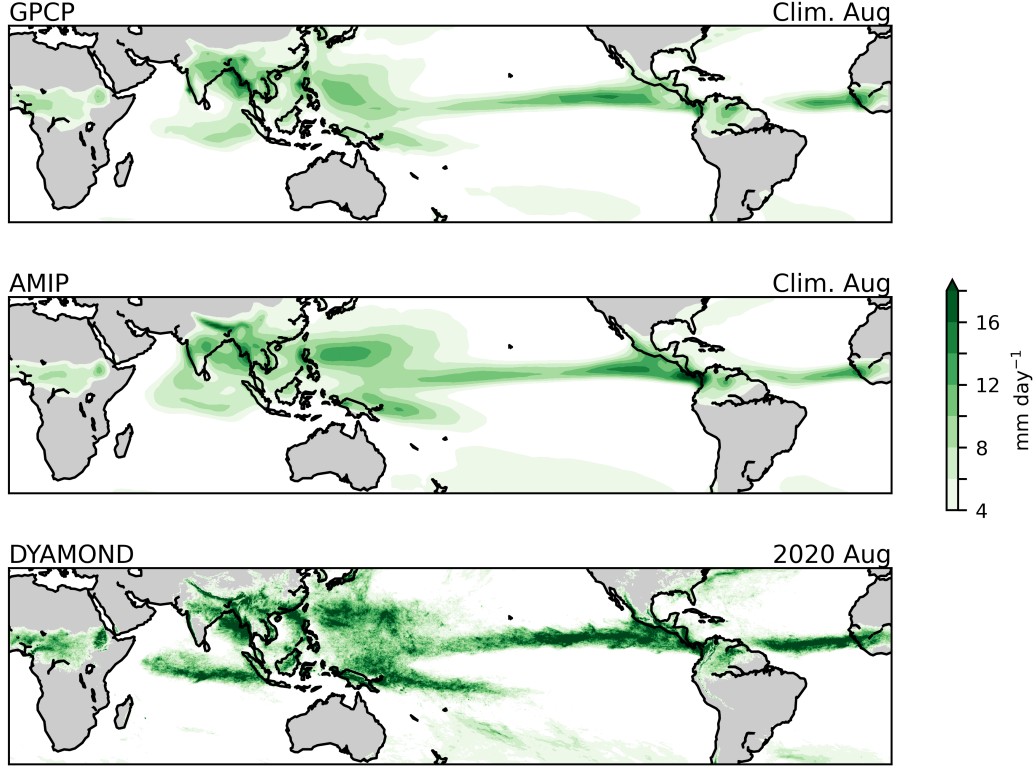

**Figure 2.** August precipitation in satellite observations (top), coarse-resolution AMIP simulations (middle), and kilometer-scale DYAMOND simulations (bottom). Observations are based on the Global Precipitation Climatology Project (GPCP) (Adler et al., 2018). AMIP simulations are from 14 models that participated in CMIP6. Both GPCP data and AMIP simulations are interpolated to a common $1° \times 1°$ grid, and the August climatology is derived from 1979–2014. The DYAMOND results, shown at the model native resolutions, are based on the average of 5 models with horizontal resolutions ranging from 3.3 km to 7.8 km for August 2020. Figure adapted from Zhou et al. (2022).

## 3 Process-based parameterizations

The uncertainties and biases in climate simulations, as shown in Figs. 1 and 2, have their roots in the parameterization of unresolved small-scale processes. So far, these processes have been primarily parameterized based on process knowledge, in a reductionist approach. For example, the influential work of Arakawa and Schubert (1974) laid the foundation for widely used parameterizations of moist convection, employing a reductionist process model of convective plumes that are, at all times, in statistical equilibrium with their environment and incorporate environmental air by entrainment. Research over the past two decades has focused on refining the formulation of the entrainment rate, a key control on climate model sensitivity to greenhouse gas concentrations (Stainforth et al., 2005; Knight et al., 2007). Typically, this rate is represented by a constant parameter $\epsilon = \theta$ or a parametric function $\epsilon = \epsilon(z, \zeta; \theta)$ of height $z$ and (usually local) plume and environmental properties encoded in the model state $\zeta$ (e.g., de Rooy et al., 2013; Yeo and Romps, 2013; Anber et al., 2019; Savre and Herzog,

2019; Cohen et al., 2020). Similarly, diffusive closures of various types have been commonly employed for boundary layer turbulence in the atmosphere and oceans. These closures employ diffusivities that may depend on height, other flow variables, or a turbulence kinetic energy determined by separate equations, and they are sometimes augmented by correction terms to represent upgradient fluxes in convective boundary layers (e.g., Mellor and Yamada, 1982; Large et al., 1994; Lock et al., 2000).

The process-based approach offers the advantage that the parameters or parametric functions that require closure are interpretable and theoretically measurable. For example, Monin-Obukhov similarity theory reduced the problem of parameterizing turbulence in a thin ($\sim 100$ m) layer near the surface to finding universal functions that characterize the vertical structure of turbulent fluxes (Foken, 2006). Later, these functions were empirically derived based on measurements over a field of wheat stubble during the summer of 1968 in Kansas (Businger et al., 1971); they have since been widely incorporated into climate models. This represents a success story for the process-based approach. It led to a parametrically sparse and interpretable representation of near-surface turbulent fluxes. It applies not only to summer conditions over Kansas wheat fields but also demonstrates invariance properties that make it applicable across most of the globe, with relatively few limitations, particularly in convective situations.

However, despite this progress, the dominant source of uncertainties and biases in climate simulations, even 50 years after the introduction of the convection parameterization by Arakawa and Schubert (1974), lies in the representation of turbulence, convection, and clouds (including their microphysics) above the near-surface layer. This indicates that the reductionist approach to developing process-based models for these components has encountered significant challenges. For example, measuring entrainment rates directly, despite being theoretically possible, remains challenging in practice, both in observational data and high-resolution simulations (e.g., Romps, 2010). The search for universal functions to accurately represent entrainment has been unsuccessful thus far. Consequently, the process-based approach to modeling convection and clouds is widely perceived as being deadlocked (Randall et al., 2003; Randall, 2013; Gentine et al., 2018).

However, prematurely dismissing process-based modeling as obsolete would ignore its advantages and its potential for further development. Contrasting the achievements of Monin-Obukhov similarity theory and moist convection parameterizations is illuminating. Monin-Obukhov similarity theory systematically coarse-grained the equations of motion, employing controlled approximations and identifying the nondimensional groups of variables that govern near-surface turbulent fluxes. The approach reduced the closure problem to finding universal functions of the identified nondimensional groups, with well-defined limits in different scenarios. This led to its near-universal applicability. In contrast, moist convection parameterizations in current use emerged phenomenologically, without a systematic coarse-graining of the known equations of motion through controlled approximations. Even when starting from a rigorous basis like the Arakawa and Schubert (1974) parameterization, operational parameterizations often introduced artificial scale breaks between boundary layer turbulence, shallow convection, and deep convection, or even between convection over land and oceans, leading to separate parameterizations with discontinuous differences in parameters and structure. Such discontinuities do not exist in nature. As a result, these parameterizations lack well-defined limits. For example, they do not converge to a well-defined dry limit when the latent heats of fusion and vaporization of water approach zero, and they do not converge to the Navier-Stokes equation as resolution increases. This approach

hindered the systematic removal of unnecessary approximations, particularly as model resolution increased and common assumptions, such as small plume area fractions relative to the host model's grid scale, or statistical equilibrium between moist convection and the environment, became inadequate (Arakawa et al., 2011; Arakawa and Wu, 2013; Randall, 2013). Therefore, rather than declaring process-based modeling for moist convection and other complex processes at a dead end, a more nuanced perspective recognizes the need for further development with greater mathematical and physical rigor, particularly in light of the abundant data and enhanced computational capabilities available today that surpass what the early pioneers of these approaches had at their disposal. The invariance properties, such as conservation laws and symmetries, inherited by this approach from the underlying equations of motion, may well hold the key to developing universal parameterizations that enable us to minimize the loss (1) for unobserved climate statistics and generalize beyond the observed distribution. That is, progress on macroscopic *techne* here hinges on microscopic *episteme*.

These considerations suggest that successful process-informed parameterizations satisfy four clear requirements:

1. Parameterizations should be grounded in the governing equations of subgrid-scale processes whenever feasible. Equations for parameterizations can be obtained by systematic coarse-graining through methods such as conditional averaging—for example, resulting in distinct equation sets for coherent structures like updrafts and their more isotropically turbulent environment—or the derivation of moment equations rooted in distribution assumptions on subgrid-scale fluctuations. Whatever approach is adopted, it is crucial that assumptions are explicitly laid out and subject to empirical validation or revision. Examples of such approaches for turbulence and convection include Lappen and Randall (2001), Golaz et al. (2002), Soares et al. (2004), Siebesma et al. (2007), Witek et al. (2011), Guo et al. (2015), Firl and Randall (2015), Tan et al. (2018), Thuburn et al. (2018), Cohen et al. (2020), and Lopez-Gomez et al. (2020).

2. Artificial scale breaks, such as those between shallow and deep convection, should be avoided. These breaks lack correspondence in nature but introduce unphysically discontinuous dynamical transitions. They also lead to correlated parameters that are difficult to calibrate with data. For example, discontinuous transitions between shallow and deep convection impede an accurate simulation of the diurnal cycle of convection (Christopoulos and Schneider, 2021; Tao et al., 2024) and result in correlated parameters in the shallow and deep convection schemes that are difficult to identify from data. Moreover, bridging these discontinuous scale breaks becomes problematic as resolution increases, for example, through the "gray zone" where processes such as deep convection become partially resolved.

3. When scale separation is absent between the parameterized subgrid-scale processes and the resolved grid-scale, parameterizations must incorporate subgrid-scale memory and stochastic terms. This implies that convection and cloud parameterizations, for example, must be explicitly time-dependent (i.e., have memory) and cannot be assumed to be in instantaneous equilibrium with the environment. Homogenization theories such as those of Mori and Zwanzig (Zwanzig, 2001), which employ averaging but also shows how fluctuations about averages arise on the macroscale, support the inclusion of these features (Majda et al., 2003; Wouters and Lucarini, 2013; Lucarini et al., 2014; Wouters et al., 2016; Lucarini and Chekroun, 2023).

4. Parameterization schemes for different processes must be coupled such that they interact consistently (Devine et al., 2006; Gross et al., 2018). For example, models for subgrid-scale fluctuations of cloud dynamics must be coupled consistently with parameterizations for cloud microphysics, ensuring that nonlinear interactions between microphysical processes such as ice nucleation and the thermodynamics and velocities of updrafts are consistently modeled (Gettelman et al., 2019). This is likely particularly important for processes such as the formation of supercooled liquid in strong updrafts, which occur out of thermodynamic equilibrium and hence are dependent on the history, and not just the instantaneous state, of air masses. Such processes are known to strongly affect the response of climate models to increased greenhouse gas concentrations (Zelinka et al., 2020).

Developing process-based parameterizations by systematically coarse-graining equations of motion will lead to unclosed terms, similar to the universal functions in Monin-Obukhov similarity theory. These terms should be expressed in terms on nondimensional variable groups that make them as "climate-invariant" as possible (Beucler et al., 2024). Whether they contain parameters, parametric functions, or non-parametric functions, they then become excellent targets for AI-enabled learning from data.

There is accumulating evidence that a program focused on process-based parameterizations that satisfy the above requirements can achieve success. For example, Lopez-Gomez et al. (2020) and Cohen et al. (2020) have demonstrated the effectiveness of a unified parameterization approach for the spectrum of small-scale motions from boundary layer turbulence to deep moist convection. This parametrically sparse approach is based on conditionally averaged equations of motion, which leads to additional evolution equations for subgrid-scale quantities such as updraft energies and mass fluxes. The equations for the subgrid-scale quantities carry additional information, including subgrid-scale memory, augmenting the information available on the grid scale of a model. Within one continuous parameterization framework, they are able to accurately represent a wide range of cloud dynamics observed on Earth, from stable boundary layers to stratocumulus-topped boundary layers and deep convection. Furthermore, Lopez-Gomez et al. (2022) have shown that machine learning can be employed to identify closure functions in these parameterizations, such as entrainment rates that depend on climate-invariant nondimensional groups.

As climate models reach resolutions where deep convection becomes marginally resolved, using an inadequate deep-convection parameterization based on instantaneous statistical equilibrium assumptions may well be less effective than not using any parameterization at all. For this reason, in the kilometer-scale simulations shown in Figures 1 and 2, deep convection parameterizations are either turned off or run with reduced activity (Stevens et al., 2019). However, parameterizations for boundary layer turbulence and low-cloud cover are usually kept, and sometimes also those for shallow convection (e.g., in IFS in Fig. 1), even though they were originally developed for resolutions in the hundred kilometer range, where, for example, assumptions of instantaneous statistical equilibrium of subgrid-scales with resolved scales are more justifiable. As seen in the above figures, this approach has not yet achieved the hoped-for success; in particular, it has not significantly improved large-scale precipitation simulations at kilometer-scale resolutions. Consequently, it is essential to advance the development of parameterizations that effectively bridge the scales between marginally resolved convection and the dynamics that remain unresolved in this resolution range, in addition to parameterizations of yet smaller scales, such as the microphysics of cloud droplet and ice crystal formation.

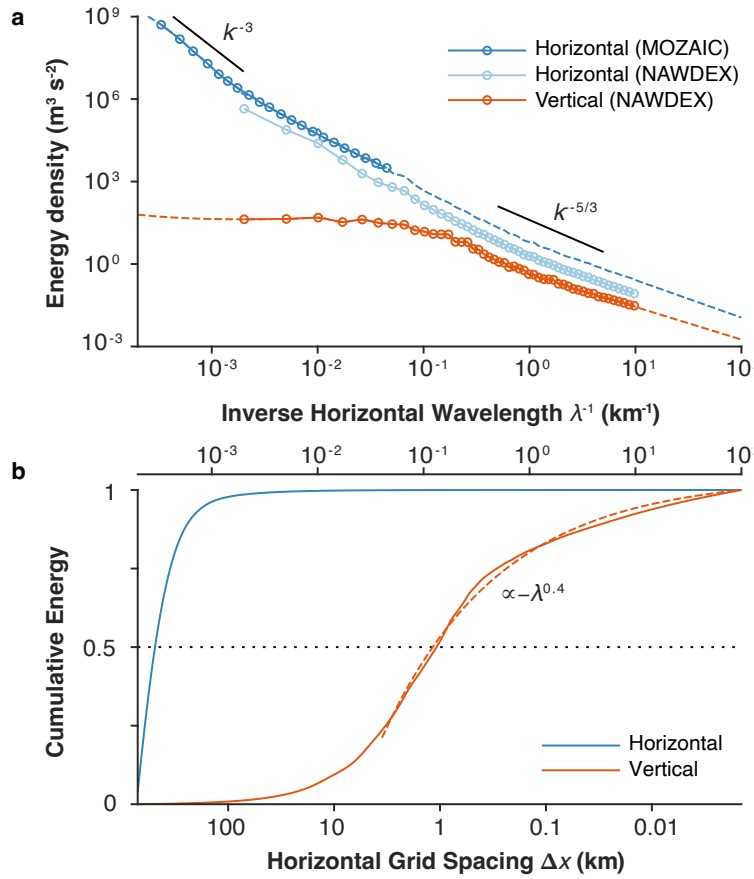

**Figure 3.** Kinetic energy spectra and cumulative energy in the atmosphere. (a) Spectral kinetic energy density based on aircraft measurements, shown as a function of inverse horizontal wavelength $\lambda^{-1} = k/(2\pi)$. (b) Cumulative kinetic energy from $\lambda_{\max} = 5000$ km to $\lambda$ on the upper horizontal axis, normalized by the energy for $\lambda_{\min} = 10$ m. The lower horizontal axis expresses wavelength $\lambda$ through the required horizontal model grid spacing $\Delta x$, using $\lambda \approx 7\Delta x$. Blue for horizontal motion kinetic energy, red for vertical motion kinetic energy. Data from Callies et al. (2014) (MOZAIC) and Schumann (2019) (NAWDEX). Dashed lines in (a) indicate linear extrapolations in log-log space, except for the dashed blue line where NAWDEX and MOZAIC data overlap: there the dashed line represents the NAWDEX spectrum multiplied by a fitting constant to match the MOZAIC spectrum. Cumulative energies are obtained by numerical integration over the spectra, including the extrapolations. The dashed line in (b), for wavelengths $\lambda \leq 25$ km, represents a power law fit $1 - a(\lambda^{\beta} - \lambda_{\min}^{\beta})$ to the cumulative vertical kinetic energy, with $\beta \approx 0.4$ and $a \approx 0.5$. Note that the extrapolations of vertical kinetic energy to large scales may not be very accurate due to possible deviations from a completely flat spectrum, which can slightly shift the position of the inflection point in the cumulative energy (Skamarock et al., 2014; Schumann, 2019).

## 4 Resolution

Climate is regulated by turbulent motions in the atmosphere and oceans. Horizontal motions transport energy, momentum, and, in the atmosphere, water vapor, shaping surface temperatures, winds, and precipitation patterns. Vertical motions couple the atmosphere and surface, creating clouds, driving precipitation, and mixing heat and tracers such as carbon dioxide in the oceans. Representing these turbulent motions accurately is crucial for climate models, but challenging due to their vast range of length scales, from planetary to millimeter scales.

Figure 3a shows the kinetic energy spectrum of horizontal and vertical motions in the atmosphere, measured by aircraft. The spectra are displayed as functions of the inverse horizontal wavelength $\lambda^{-1}$, which is proportional to the horizontal wavenumber $k = 2\pi/\lambda$. At large scales (small wavenumbers), the spectrum of horizontal kinetic energy follows a $k^{-3}$ power law, as predicted by geostrophic turbulence theory (Vallis, 2006, chapter 9). At mesoscales below approximately 500 km, the spectrum becomes shallower, resembling a $k^{-5/3}$ power law. The reason for this change has been debated. The shallower spectrum seems to be caused by linear inertia-gravity waves, which are internal waves modified by planetary rotation that coexist with the nonlinear, primarily geostrophic, atmospheric turbulence (Dewan, 1979; VanZandt, 1982; Callies et al., 2014).

At scales greater than 10–20 km, the kinetic energy of vertical motions is much weaker than that of horizontal motions, with a relatively flat spectrum. This difference is mainly due to two factors: (1) The scale of vertical motions is limited by the depth of the troposphere (about 10–20 km), which contains the most important vertical motions; and (2) the vertical velocity depends on the divergence of the horizontal velocity, which is weaker (by a factor of order Rossby number) than the dominant rotational velocity at large scales, though it becomes comparable to it on mesoscales. The divergence involves horizontal derivatives, leading to a multiplication by $k^2$ of the kinetic energy spectra in wavenumber space at horizontal scales above 10–20 km, where the vertical depth scale is constrained by the depth of the troposphere (see Schumann (2019) for a detailed model, from which these insights are drawn). This results in the relatively flat spectrum with low vertical kinetic energy at larger horizontal scales. At horizontal scales smaller than about 10–20 km, where the horizontal scale is comparable to the vertical scale and the latter is no longer constrained by the depth of the troposphere, the vertical kinetic energy spectrum starts to decay following a rate of roughly $k^{-5/3}$, like the horizontal kinetic energy spectrum. At yet smaller scales in the meter range, the turbulence becomes increasingly isotropic, which also results in a $k^{-5/3}$ power law because three-dimensional turbulence follows a Kolmogorov spectrum. The figure shows an extrapolation of both the horizontal and vertical kinetic energy spectra from the smallest measured scale near 100 m down to 10 m for illustrative purposes. However, in reality, the spectra continue without a break to the Kolmogorov scale at millimeters, where kinetic energy is dissipated.

As horizontal climate model resolution increases, the continuity of the atmospheric energy spectrum implies a gradual improvement as resolved motions replace imperfectly parameterized smaller scales. To quantitatively assess the benefits of higher resolution in climate models, we integrate the energy spectra $\hat{E}(k)$ over a wavenumber interval from $k_{\min} = 2\pi/\lambda_{\max}$ to $k = 2\pi/\lambda$:

$$E(k_{\min}, k) \propto \int_{k_{\min}}^{k} \hat{E}(k')\,dk'. \tag{2}$$

Figure 3b illustrates the cumulative energy contained between $\lambda_{\max} = 5000$ km and a given $\lambda$ on the upper horizontal axis, normalized by the cumulative energy extrapolated to $\lambda_{\min} = 10$ m. Because of the steepness of the horizontal kinetic energy spectrum at large scales, the benefits of increased resolution for horizontal kinetic energy level off at wavelengths just under 1000 km. This corresponds to a grid spacing around $\Delta x \approx 150$ km (lower horizontal axis in Fig. 3) because the minimum wavelength $\lambda$ a model can resolve is approximately $7\Delta x$ (Skamarock, 2004; Wedi, 2014; Klaver et al., 2020). Climate models reached this "geostrophic turbulence plateau" in resolution in the past decade (Schneider et al., 2017b). However, the vertical kinetic energy spectrum remains relatively flat at larger scales, leading to continued benefits in resolving vertical kinetic energy as $\lambda$ decreases.[3] Concretely, the data in Fig. 3 indicate that resolving wavelengths of 1000 km, 100 km, and 10 km (grid spacings $\Delta x$ of about 150 km, 15 km, and 1.5 km, respectively) increases the fraction of resolved vertical kinetic energy between 5000 km and 10 m from 0.6% to 7% and 43%, respectively. The returns on increasing resolution only begin to diminish for wavelengths below 1 km, that is, grid spacings $\Delta x \lesssim 150$ m. The specific results depend on the data and extrapolations used (Schumann, 2019), but the main finding is clear: Even at kilometer-scale resolution, most vertical motions require parameterization. This is especially true for the motions at horizontal scales in the meter to hundred meter range, which generate the low clouds that help control Earth's energy balance (Bony and Dufresne, 2005; Stephens, 2005; Vial et al., 2013; Schneider et al., 2017b, 2019). Therefore, as we push the resolution frontier, it is crucial to concurrently improve parameterization of smaller-scale turbulent motions in ways that, as discussed in section 3, are commensurate with the model resolution; for example, statistical equilibrium assumptions for subgrid-scale fluctuations must be relaxed at resolutions where scale separation between resolved and parameterized processes disappears.

Increasing the horizontal resolution of climate models incurs a substantial computational cost, which grows as $(\Delta x)^{-3}$ for a fixed vertical resolution. This cost arises from the increasing number of horizontal grid points, $\propto (\Delta x)^{-2}$, and the necessity for smaller timesteps, $\propto (\Delta x)^{-1}$, to maintain numerical stability. To illustrate using the previous example from Fig. 3, reducing the grid spacing from $\Delta x \approx 15$ km to $\Delta x \approx 1.5$ km increases the computational cost by a factor 1000, while enhancing the cumulative vertical kinetic energy resolved by a factor 6, from 7% to 43%. That is, the rate of improvement in resolved vertical kinetic energy (a low power of $\Delta x$) is significantly smaller than the additional computational expense. From this perspective, increasing horizontal resolution is an inefficient means of improving climate models. Moreover, the vertical grid spacing $\Delta z$ must also be considered and typically should scale with $\Delta x$ (Lindzen and Fox-Rabinovitz, 1989); however, existing process-based parameterizations are often manually calibrated to a specific vertical resolution, resulting in a reluctance to increase vertical resolution alongside horizontal resolution in practice. Increasing vertical resolution entails a more modest computational cost, generally scaling as $(\Delta z)^{-1}$. This is because fast vertical dynamics are generally treated implicitly in climate models to circumvent timestep limitations, ideally using implicit solvers with computational costs linear in $(\Delta z)^{-1}$. As suggested in Fig. 1, increasing vertical resolution at coarser horizontal resolution can be advantageous because it can improve the representation of parameterized subgrid-scale dynamics (Harlaß et al., 2015; Kawai et al., 2019; Smalley

---

[3]For a $k^{-\alpha}$ spectrum, the cumulative vertical kinetic energy scales as $-\lambda^{\beta}$, where $\beta = \alpha - 1$. This gives $\beta = 2/3$ for $\alpha = 5/3$. The curves in Fig. 3, for wavelengths $\lambda < 25$ km, are fit well with $\beta \approx 0.4$. This shows that there is no qualitative change in behavior over those scales, only gradual gains from increasing resolution.

et al., 2023). At higher horizontal resolutions where resolved dynamics become more isotropic, proportionately increasing both vertical and horizontal resolution becomes necessary, leading to a computational cost that scales even less favorably, like $(\Delta x)^{-4}$.

Therefore, optimizing the parameters $\lambda$ that define model resolution requires inevitable trade-offs. Even at foreseeable future resolutions, unresolved scales of atmosphere and ocean turbulence, plus even finer scales controlling cloud microphysics and other processes, will still require parameterization. While increasing resolution gradually improves the representation of turbulent dynamics and enhances the resolution of surface topography, gravity waves, and land-sea contrasts, the 1000-fold increase in computational cost from $O(10\,\mathrm{km})$ to $O(1\,\mathrm{km})$ is unlikely to justify the benefits (Wedi et al., 2020). It will remain crucial to make parameterizations as resolution-independent ("scale aware") as possible and to allocate some computational resources to calibrating parameterizations with data, which requires hundreds to thousands of climate simulations. Although calibrating over shorter (weather) timescales is computationally feasible and may be beneficial, it does not guarantee improved simulations of longer-term climate statistics (Schirber et al., 2013). Moreover, to quantify climate risks, it is necessary to run ensembles of climate simulations to broadly explore possible climate outcomes. Doing so requires $O(10)$ ensemble members to sample atmospheric and oceanic internal variability (Deser et al., 2020; Wills et al., 2020; Bevacqua et al., 2023), ideally with each of those also sampling model uncertainties by drawing from a posterior distribution over plausible models (Dunbar et al., 2021; Howland et al., 2022), resulting in hundreds of decades-long climate simulations. Ensemble generation lends itself well to distributed (cloud) computing as it is embarrassingly parallel. However, it also constrains the routinely achievable resolution in climate models. Therefore, kilometer-scale resolution remains an experimental frontier. Currently, routinely achievable atmospheric resolution lies in the 10–50 km range (Schneider et al., 2023), while ocean resolutions of 5–10 km are achievable (Chang et al., 2020; Silvestri et al., 2024). By finding the right balance between resolution and parameterization learning and calibration, we can make significant strides in improving climate simulations within realistic computational constraints.

## 5  AI for learning parameterizations

Even at the highest resolutions achievable in climate modeling, parameterizing small-scale processes remains essential. While it may be tempting to learn about all small-scale processes holistically from data, this approach is more likely to be successful in weather prediction, where short-term accuracy is prioritized, and energy conservation is less critical because daily data assimilation prevents model drift. In contrast, climate prediction faces two primary challenges.

First, energy conservation and predicting changes in Earth's energy balance are paramount, as exemplified by Suki Manabe's Nobel-prize winning work. His climate modeling work began with radiative-convective equilibrium models to explore the energetic effects of changes in atmospheric composition on the atmosphere and surface (Manabe and Strickler, 1964; Manabe and Wetherald, 1967). In the same vein, climate models must accurately predict responses to changes in atmospheric composition, such as greenhouse gas concentrations and aerosol loading. Some of these responses involve rapid adjustments that are independent of surface temperature changes. For example, changes in greenhouse gas concentrations can modulate cloud cover through rapid adjustments mediated by changes in longwave radiative fluxes, in addition to the cloud response to surface

temperature changes (Gregory and Webb, 2008; Sherwood et al., 2015; Bretherton, 2015). Predicting these effects separately is essential for future climate scenarios: in future climates, changes in aerosol loadings and greenhouse gas concentrations can decorrelate, and changes in longwave radiative fluxes can decouple from surface temperature changes, as is the case in solar geoengineering scenarios (Schneider et al., 2020). Learning their compound effects holistically from data will not enable such predictions because the effects cannot be disentangled from data alone, where, for example, changes in greenhouse gas concentrations and surface temperature are correlated; instead, modeling the processes separately using known physics as guardrails appears essential.

Second, climate change prediction is an out-of-distribution challenge, as we lack data for future, warmer climates. While using simulated data for learning is one approach, it restricts learning to model emulation and may not capture complex processes such as aerosol effects on clouds, which currently cannot be simulated reliably even in limited domains. Therefore, learning from observations must be informed by known physics to ensure models generalize beyond the observed climate distribution.

To make progress and generalize beyond observed climate data, we can build on process-based parameterizations, which encode known physics through conservation laws and invariance properties. AI-based methods, broadly understood to include methods from Bayesian to deep learning, can aid in learning about entire parameterizations for individual processes or unclosed terms and functions within them. This can reduce inaccuracies in climate models and potentially also allow the quantification of uncertainties.

AI approaches require the specification of a loss function. The most suitable loss function (1) penalizes differences between simulated and observed climate statistics, weighted by the inverse of a covariance matrix representing noise sources such as observational error and internal variability. The loss function should include variables such as TOA radiative energy fluxes and global precipitation fields, as shown in Fig. 1. It may also include higher-order statistics, such as the covariance between surface temperature and cloud cover (Schneider et al., 2017a). This covariance represents an emergent constraint: a statistic that, across climate models, correlates with the response of cloud cover to greenhouse gas concentration increases (e.g., Klein and Hall, 2015; Brient and Schneider, 2016; Caldwell et al., 2018; Hall et al., 2019). Such emergent constraints can arise, for example, from fluctuation-dissipation theorems that relate fluctuations in a system to the response of the system to external perturbations (Ruelle, 1998; Lucarini and Chekroun, 2023). Emergent constraint statistics, previously used only for retrospective model assessments, can be proactively minimized in the loss function to improve model accuracy in simulating greenhouse gas responses.[4]

However, using climate statistics in a loss function challenges traditional machine learning (ML) methods. Supervised learning (SL), the dominant ML approach, depends on labeled input-output pairs for process modeling and learns regressions of outputs onto inputs. For example, a convection parameterization requires at least temperature and humidity inputs, which must

---

[4]If emergent constraint statistics are used during loss minimization, they can no longer serve as retrospective constraints on the response of the model to perturbations. In retrospective studies, there is a risk in using emergent constraints because the correlation between emergent constraint statistics and the climate response may be spurious (Caldwell et al., 2014, 2018). When using emergent constraint statistics during loss minimization, by contrast, the statistics at worst may merely be uninformative about model parameters and processes.

be paired with the output—convective time tendencies of temperature and humidity—for training. Since such data are unavailable from Earth observations, SL has been limited to simulated data (e.g., O'Gorman and Dwyer, 2018; Rasp et al., 2018; Gentine et al., 2018; Yuval and O'Gorman, 2020; Yuval et al., 2021; Yu et al., 2023). Conversely, the climate statistics in the loss function (1) provide only indirect information about processes such as convection. For example, the loss function may include fields such as precipitation and cloud cover—noisy fields with missing data that are influenced by multiple processes, including but not limited to convection (Schneider et al., 2017a).

To illustrate, consider determining closures in a conservation equation:

$$\frac{Dq}{Dt} = \mathcal{F} + \mathcal{S}. \tag{3}$$

Here, $q(\boldsymbol{x}, t)$ is a tracer, such as total specific humidity, dependent on space $\boldsymbol{x}$ and time $t$, and $D/Dt = \partial/\partial t + \boldsymbol{u} \cdot \nabla$ is the material derivative with fluid velocity $\boldsymbol{u}(\boldsymbol{x}, t)$. The quantities on the left-hand side are taken to be resolved on the model's grid. The right-hand side consists of two components: $\mathcal{F}(\boldsymbol{x}, t)$ represents unresolved subgrid-scale flux divergences in need of parameterization; $\mathcal{S}(\boldsymbol{x}, t)$ denotes all other, separately modeled sources and sinks.

In SL approaches, the objective is to map the model state $\boldsymbol{\zeta}$ to approximate subgrid-scale flux divergences $\hat{\mathcal{F}}(\boldsymbol{\zeta}; \boldsymbol{\nu})$, with parameters $\boldsymbol{\nu}$ (e.g., neural network weights and biases). To make the problem tractable, the mapping is usually considered locally in the horizontal, mapping column states $\boldsymbol{\zeta}(z_i, t)$ at discrete levels $z_i$ (for $i = 1, \ldots, N_z$) to parameterized flux divergences $\hat{\mathcal{F}}_j(\boldsymbol{\zeta}(z_i, t); \boldsymbol{\nu}))$ at levels $z_j$. This is achieved by using the column state $\boldsymbol{\zeta}(z_i, t)$ as input, and the remainder $Dq/Dt - \mathcal{S}$ as output, to learn a regression $\hat{\mathcal{F}} \approx Dq/Dt - \mathcal{S} + \epsilon$. The material derivative $Dq/Dt$ is rolled out over time intervals typically spanning hours to days. The aim is to minimize the residual $\epsilon$ over parameters $\boldsymbol{\nu}$, typically using methods such as backpropagation that compute gradients of a loss function with respect to the parameters $\boldsymbol{\nu}$.

This approach leverages the expressive capabilities of deep learning and has shown some promise, as evidenced in studies demonstrating that moist convection or ocean turbulence parameterizations can be effectively learned in this manner (e.g., O'Gorman and Dwyer, 2018; Rasp et al., 2018; Gentine et al., 2018; Bolton and Zanna, 2019; Yuval and O'Gorman, 2020; Yuval et al., 2021; Zanna and Bolton, 2020; Wang et al., 2022; Sane et al., 2023). Error corrections to existing parameterizations have been learned in a similar manner (Watt-Meyer et al., 2021; Bretherton et al., 2022). However, focusing solely on minimizing the short-term residual $\epsilon$ presents several limitations:

1. The learned parameterization $\hat{\mathcal{F}}$ may not necessarily minimize the climate-relevant loss function (1), which is concerned with longer-term statistics, as opposed to the shorter-term trajectories of states on which SL approaches have focused. Schirber et al. (2013) provide an example of how short-term optimization over trajectories can lead to no improvement or even degradation of longer-term climate statistics in a model.

2. Supervised learning of the parameterization $\hat{\mathcal{F}}$ is typically restricted to data generated computationally in higher-resolution simulations, restricting it to the emulation of imperfect models, because labeled parameterization output $Dq/Dt - \mathcal{S}$ is generally not available from Earth observations.

3. The parameterization $\hat{\mathcal{F}}$, typically learned for a multitude of processes jointly, usually does not generalize well out of the training distribution and is resolution dependent, necessitating training with a broad range of simulated climates (e.g., O'Gorman and Dwyer, 2018) and re-training whenever the resolution is changed.

4. Climate models incorporating the learned parameterization $\hat{\mathcal{F}}$ often struggle with conserving essential quantities such as energy and exhibit instabilities during extended integrations (e.g., Brenowitz and Bretherton, 2018), because minimizing the short-term residual $\epsilon$ does not inherently ensure conservation or stability.

Some of the challenges associated with SL approaches in climate modeling can be addressed or alleviated. For example, longer roll-outs of the material derivative $Dq/Dt$ have been shown to reduce instabilities when integrating the learned parameterization $\hat{\mathcal{F}}$ into a climate model (Brenowitz et al., 2020), and constraints on the loss function may be used to enforce conservation laws (Beucler et al., 2021). Additionally, the issue of resolution dependence in the learned parameterization $\hat{\mathcal{F}}$ can be tackled by shifting from learning a finite-dimensional discrete mapping between model grid points to learning an infinite-dimensional operator. Such operators map between function spaces; they would effectively represent the atmospheric or oceanic column state as a continuous function, rather than as a set of discrete points (e.g., Nelsen and Stuart, 2021; Kovachki et al., 2023). This approach allows for a more flexible representation of the underlying physical processes, potentially adaptable to different resolutions without the need for retraining.

An alternative approach that avoids the restrictions of SL views learning parameterizations $\hat{\mathcal{F}}$ and ML parameters $\boldsymbol{\nu}$ within them as an inverse problem (Kovachki and Stuart, 2019), minimizing a climate-relevant loss function (1) that focuses on statistics. However, this loss function is based on data that are only indirectly informative about the process being modeled; that is, the parameterization $\hat{\mathcal{F}}$ influences the climate model output $\boldsymbol{\mathcal{G}}(t; \boldsymbol{\theta}, \boldsymbol{\lambda}, \boldsymbol{\nu}; \boldsymbol{\zeta}_0)$ in the loss function only indirectly, through the complex and nonlinear interactions of other components in the climate model (Schneider et al., 2017a). Gradients of the loss function with respect to the parameters $\boldsymbol{\nu}$ in this case would involve differentiation through the model $\boldsymbol{\mathcal{G}}$, which may not be differentiable (e.g., at discontinuous phase transitions) or may be difficult to differentiate.

In this context, learning about the parameterization $\hat{\mathcal{F}}$ is no longer a straightforward regression of outputs onto inputs. However, this does not preclude the inclusion of parameter-rich and expressive deep learning models within the parameterization. The parameters $\boldsymbol{\nu}$, together with process-model parameters $\boldsymbol{\theta}$, can be estimated by minimizing a climate-relevant loss function using derivative-free ensemble Kalman inversion techniques, which are proven to scale well to high-dimensional problems, can be used with models that are not or difficult to differentiate, and have smoothing properties that are desirable for chaotic dynamical systems such as the climate system (Kovachki and Stuart, 2019). As in many inverse problems, minimizing the loss function is often an ill-posed problem with many possible solutions, which may be sensitive to small changes in the data (Tarantola, 1987; Hansen, 1998; Iglesias et al., 2013). This requires regularization, for example, through the use of prior information on the parameters $\boldsymbol{\nu}$ to select "good" parameter sets among the many that may minimize the loss. Such prior information may be obtained, for example, by pre-training on computationally generated data, which can be more detailed than observational data (Lopez-Gomez et al., 2022). This inverse problem approach, augmented with prior information, not only makes it possible to learn from heterogeneous and noisy Earth observations but also allows for the quantification of uncertainties (e.g.,

Cleary et al., 2021; Huang et al., 2022). Stochastic elements can also be incorporated in the parameterizations (e.g., Schneider et al., 2021b), which, as discussed in section 3, is particularly relevant in the absence of clear scale separation, offering a more principled and realistic representation of climate processes.

This expanded perspective on incorporating AI methods in climate models broadens the scope of where these methods can be effectively integrated. It moves beyond automatically calibrating scalar parameters in climate models (Zhang et al., 2015; Couvreux et al., 2021; Hourdin et al., 2023) to encompass higher-dimensional parameter spaces, including those relevant to deep learning approaches. Rather than focusing solely on areas where SL is feasible, the emphasis shifts to where AI can have the most significant impact. The key challenge in climate modeling and prediction is minimizing the loss function (1) for unobserved climate statistics, especially in global warming scenarios where these statistics may fall outside the range of observed data. While traditional methods such as withholding part of the data for cross-validation remain essential, they fall short in ensuring model generalization beyond the training dataset. This limitation becomes evident when attempting tasks such as predicting rapid adjustments in cloud cover (Gregory and Webb, 2008; Sherwood et al., 2015; Bretherton, 2015) or changes in the photosynthetic productivity of the biosphere (Luo, 2007) in response to increased $CO_2$ concentrations, given the limited range of $CO_2$ concentration variations and their inextricable correlation with temperature variations in recent observations. Embedding AI-driven closures within process-based parameterizations rooted in conservation laws can help in obtaining models that generalize out of the observed climate distribution.

A valuable insight emerges from the success of similarity theories, such as the Monin-Obukhov similarity theory discussed in section 3, which generalized effectively from a few specific measurements to a wide range of global conditions. Similarly, AI methods may be most effective when used to learn universal functions of relevant nondimensional variable groups: functions that likely remain invariant across different climates and are well-sampled in current climate conditions, including the seasonal cycle whose amplitude in many quantities exceeds the climate change signals we expect for the coming decades (Schneider et al., 2021a). If such learning minimizes a loss (1) in an online setting, that is, while the learned functions are integrated into the large-scale model, it is more likely to lead to long-term stable models, because existence of the statistics in the loss function implies that the model is stable over the timescales over which the statistics are aggregated. For example, rather than learning the convective flux divergence $\hat{\mathcal{F}}$ for water or energy holistically, it is likely beneficial to focus on learning corrections to process-based parameterizations or key unknown functions such as entrainment and detrainment rates within the coarse-grained conservation laws for water and energy, embedded online in a larger forward model. Lopez-Gomez et al. (2022) demonstrated that this approach can successfully learn parameterizations that generalize well to warmer climates not encountered during training; related results, with gradient-based online learning approaches, are emerging for turbulence closure models (Shankar et al., 2024). An ancillary benefit is that quantification of uncertainties becomes more straightforward, and the parameterizations remain interpretable, facilitating the investigation of mechanisms, for example, of cloud feedbacks and the differential effects of changing greenhouse gas concentrations and warming in them. Models for structural errors can similarly be incorporated where the errors are actually made—within the parameterizations of unresolvable small-scale processes (Levine and Stuart, 2022; Wu et al., 2024), rather than in the space of the model output (Kennedy and O'Hagan, 2001).

Therefore, we advocate for an approach that leverages our extensive knowledge of conservation laws, expressed as partial differential equations, and enhances it with AI methods to learn about closure functions in parameterizations where reductionist first-principle approaches fall short. The central challenge is to find a balance: using first principles to encode system knowledge and conservation laws for generalization and interpretability, while avoiding overly rigid constraints that limit the model's adaptability to diverse data sets. This balance will vary across different components of the climate system, and finding it requires

domain expertise. For instance, first principle modeling has proven less effective than data-driven approaches for river flows and snowpack thickness, where systematic coarse-graining is challenging and observed spatial and temporal variations plausibly sample future scenarios (Kratzert et al., 2018, 2019; Nearing et al., 2021; Kratzert et al., 2023; Moshe et al., 2020; Charbonneau et al., 2023; Nearing et al., 2024). In contrast, modeling phenomena such as turbulence, convection, and clouds, including their microphysics, may benefit more from reductionist process-informed modeling, among other reasons because rapid radiative

adjustments impact clouds and cannot be learned solely from data, but spatial and temporal variations in data may sample climate-invariant closure functions appearing in them well. High-resolution simulations, for example, of the dynamics of clouds and convection (Hourdin et al., 2021; Shen et al., 2022) or of ocean turbulence (Wagner et al., 2024), Lagrangian particle-based simulations of cloud microphysics (Shima et al., 2009; Morrison et al., 2020; Azimi et al., 2024), and laboratory data provide increasingly rich datasets for pre-training of individual parameterizations and closure functions within them. Pre-training can

occur in a compartmentalized manner, process-by-process, before jointly fine-tuning all processes with observations. Striking the right balance between first-principle and data-driven approaches is crucial for developing climate models that are physically grounded and trustworthy for predictions beyond observed climates, yet flexible enough to integrate a wide range of data, leading to more accurate and reliable predictions.

## 6  A balanced path forward

Climate models, as a form of *techne*, aim to provide the most accurate and reliable predictions of how the climate system's statistics will change under unobserved conditions, such as increased greenhouse gas concentrations or changes in aerosol loadings. Improving climate models is urgent for proactive and effective adaptation to the coming climate changes. However, current models fall short in accuracy and reliability (Fiedler et al., 2021; President's Council of Advisors on Science and Technology, 2023), as evidenced by their still significant errors in simulating observed climate statistics (Fig. 1).

Progress in climate modeling has been gradual, achieved primarily through increasing resolution and refining process-based parameterizations for small, unresolvable scales. Yet, neither approach alone, nor in combination, seems likely to produce a significant leap in model accuracy and reliability. The complexity of the climate system limits the effectiveness of reductionist approaches in developing process-based parameterizations. Additionally, while higher resolution is beneficial, it is no panacea. At any resolution reachable in the foreseeable future, many aspects, such as large portions of the atmosphere's and oceans'

vertical motion and finer scales such as those controlling cloud microphysics, will remain unresolvable.

AI tools, in their broadest sense, hold promise for breakthroughs due to their capacity to learn from high-resolution simulations and from the extensive array of Earth observations available. However, they cannot operate in isolation. Climate change

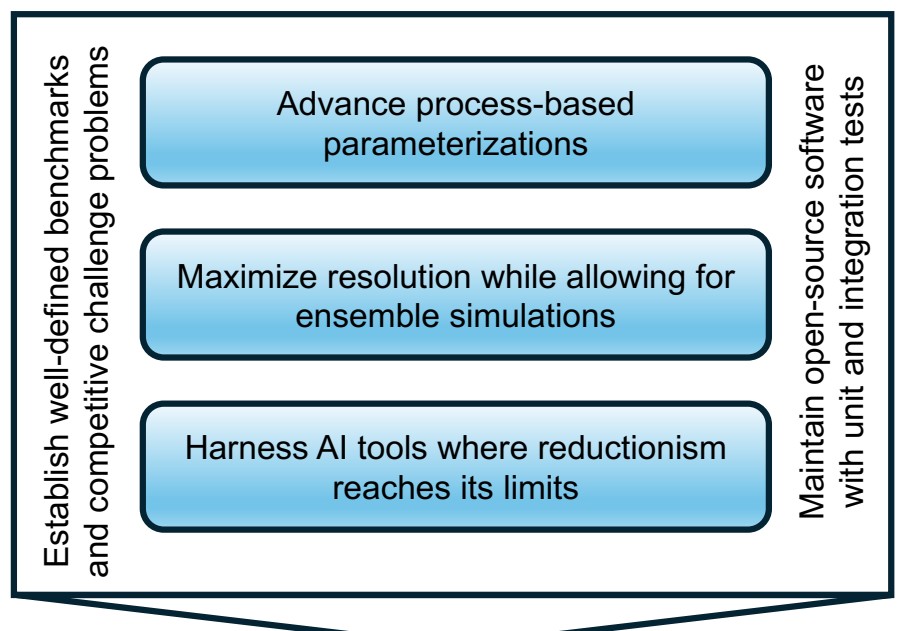

**Figure 4.** A balanced approach capitalizing on all tools at our disposal is the most promising path toward accurate, interpretable, and trustworthy climate predictions and projections that can inform decision making.

prediction is an archetypal out-of-distribution prediction challenge. It is difficult to envision how an unsupervised AI system could learn the effects of unseen greenhouse gas concentrations on aspects such as cloud cover or the biosphere's photosynthetic productivity using only higher-resolution simulations and current and recent past observations. The limited range of greenhouse gas variations is closely correlated with temperature changes in recent observations, making it challenging to isolate their individual effects, which need to be predicted. Unlike weather forecasting, where short-term predictions can be validated daily and long-term stability and conservation properties of simulations are less critical, climate prediction lacks the luxury of immediate validation. Conservation, long-term stability in an "infinite forecast," and reliable generalization beyond observed climate states are essential. Trust in climate predictions and the absence of immediate validation additionally require models to be interpretable and uncertainties in predictions to be quantified.

Therefore, a balanced approach that capitalizes on the strengths of all three dimensions—advancing process-based parameterizations, maximizing resolution while allowing ensembles of simulations, and harnessing AI tools to incorporate data-driven models where reductionism reaches its limits—in our view is the most promising path forward (Schneider et al., 2021a) (Fig. 4). In situations where we have well-defined equations of motion and can systematically coarse-grain them, AI may be optimally employed to learn data-driven yet climate-invariant closure functions of nondimensional variable groups arising

within coarse-grained equations. This approach is akin to how data have been used to close Monin-Obukhov similarity theory for the atmospheric surface layer. Conversely, in situations where first-principle modeling and systematic coarse-graining are less effective, but where spatial and temporal climate variations—particularly the seasonal cycle—may plausibly represent future climate states, more direct data-driven models could prove more fruitful. This may be particularly relevant for various aspects of land surface modeling, such as snow, vegetation, and river models.

To catalyze advances in climate modeling, we advocate for the establishment of well-defined competitive challenge problems, employing open benchmark data, shared code, and clear quantitative success metrics. Such challenge problems can foster innovation in climate process modeling, as they have in other areas, such as machine vision, natural language processing, and protein folding (Donoho, 2023). For example, benchmark challenges for cloud parameterizations could leverage libraries of high-resolution simulations, employing current-climate simulations for training and altered climate conditions for evaluation (e.g., Hourdin et al., 2021; Shen et al., 2022; Lopez-Gomez et al., 2022; Yu et al., 2023). Other benchmark challenges may focus on the seasonal cycle of land carbon uptake, evapotranspiration, snow cover, or river discharge, using a subset of the available data for model training while reserving other regional datasets for evaluation. Benchmarking can also include retrospective analysis of emergent properties such as climate trends in historical simulations, as long as the metrics evaluated were not used in model calibration. Designing such structured challenges can drive innovation, help determine what balance between process-based and data-driven methods is most successful, and lead to more accurate and reliable climate models.

Moreover, to engender trust in climate predictions, it is imperative to develop and maintain carefully designed open-source software, accompanied by rigorous unit and integration tests. This approach ensures transparency, reproducibility, and replicability (National Academies of Sciences, Engineering, and Medicine, 2019), enabling the scientific community and stakeholders to scrutinize and validate the models' predictions. Trustworthy software infrastructure is a cornerstone for building confidence in climate models and their predictions, especially as we integrate more complex data-driven components into modeling frameworks.

Ultimately, the utility of climate predictions for decision-making hinges on their trustworthiness and their ability to explore a broad range of possible climate outcomes through ensembles (Deser et al., 2020; Bevacqua et al., 2023). A balanced approach that is grounded in decades of accumulated intellectual capital, rigorous approximations, and carefully designed software is likely to foster such trust. This approach can enable a clear tracing of the causal chain leading to possible climate changes, allowing for interpretation and scrutiny in line with centuries-old scientific traditions. If successful, this strategy may eventually also narrow the gap between *episteme* and *techne* in climate modeling. It may deepen our understanding of the climate system's complexities to investigate models that integrate data-driven components and to use them to shed light on very different past climates, such as at the Last Glacial Maximum or during the Eocene hothouse climates. Such a convergence would mark a significant advance in both the science and practical application of climate modeling.

*Code and data availability.* The data and code needed to produce Fig. 1 is available at doi.org/10.22002/z24s9-nqc90, and those needed to produce Fig. 3 are available at doi.org/10.22002/qemqk-rgq45.

*Author contributions.* Tapio Schneider: Conceptualization, formal analysis, visualization, writing (original draft); Ruby Leung: Conceptualization, visualization, writing (editing and review); Robert Wills: formal analysis, visualization, writing (editing and review).

*Competing interests.* The authors declare no competing interests.

*Acknowledgements.* We thank Joern Callies and Ulrich Schumann for providing the data for Figure 3 and for their valuable discussions; Peter Caldwell, Yi-Fan Chen, Raffaele Ferrari, Nadir Jeevanjee, Thomas Müller, Fei Sha, and R. Saravanan for insightful comments on drafts; and Duan-Heng Chang for identifying a critical typo in a previous version of the analysis script for Figure 1. The research on which this essay draws is supported by Schmidt Sciences, the U.S. National Science Foundation (grant AGS-1835860), and the Swiss National Science Foundation (Award PCEFP2_203376).

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
