# Peer review of "Opinion: Optimizing climate models with process-knowledge, resolution, and AI"

_EGUsphere, 2024_

## Referee Comment (RC1)

Review of Schneider et al 2024 Optimizing climate models

I liked this article a lot.

1. I believe that you shouldn't trust a climate model prediction that you don't understand conceptually. This is particularly necessary for climate modeling because we can't validate any of our predictions until it is too late. Accepting anything on blind faith from a black box model seems like a recipe for disaster. This requires a convergence between episteme and techne which is different from your framing around line 25. Regarding line 248, Bjorn has told me that not using deep convection is strongly motivated by a desire to understand what his model is doing rather than just because it makes the simulation better (which I think most km-scale modelers at this point believe is not necessarily true)

2. I felt that the assessment of km-scale models on p. 7 (and, to a lesser extent, high-res models on p.6) was a bit unfair. My feeling is that conventional GCMs have been optimized and tuned for decades but these higher-resolution analogues are still new and generally haven't been well tuned. I think they have a lot of room for improvement. It is hard to say at this point how much benefit they will provide, but they will certainly be better than the versions you're analyzing. One particular challenge is that they are too expensive to tune, which both supports my claim that they have a lot of potential for improvement but also your point that km-scale models may be too expensive to be practical.

3. Paragraph between line 190 and 195: my personal feeling is even stronger than your argument here – I don't think it's clear that it will ever be possible to adequately parameterize clouds from variables available on the grid scale. Necessary information may simply not be available. I can't think of how to edit your text to express this, so just adding as a comment.

4. L221: When you say "conditional averaging", I think you mean that you will break terms in the governing equations into summands satisfying one condition or another. Just averaging over one particular condition (e.g. only for updrafts) generally does not result in a statement equivalent to the original governing equations. I've had postdocs go astray this way.

5. Numerated item beginning on L220: I would add that carefully applying scaling arguments to make simplifications and being explicit about the simplifications you're making is critical for readers to understand what you're doing and to be able to assess how much trust they should have in what you do. It may turn out that some assumption you make (like the PDF for subgrid variability) turns out to be inappropriate in some edge case and having those assumptions be clearly listed will help in tracking down these issues.

6. To amplify the last comment, I believe that assumed PDF shape and in particular assumed covariances between variables will be central to the skill of the kind of model you're advocating.

7. Discussing surrounding the list of desired properties for parameterizations starting around L219: I think you're missing the possibility for covariance between variables within processes and particularly sub-grid scale interactions between processes. For example, condensation is nonlinearly stronger in portions of a grid cell with stronger updrafts, which wouldn't be captured in models where condensation is performed in microphysics rather than turbulence schemes. I also like Devine et al (2006; GRL), which points out that

interactions between convective transport and sub-grid scale spatial variations in DMS are critical for getting cloud microphysics right. All these covariances between processes are things that improved resolution fixes, but would be hard to parameterize without having a single really complicated parameterization that does everything.

8. Paragraph starting on L304: It is interesting that most modeling centers have found that decreasing dx provides better simulation skill but decreasing dz generally makes the model worse (at least without a ton of extra work). I think this is because the model is actually *more* sensitive to vertical grid and gets its skill from tuning rather than accurate discretized equations, and because it is easy to make discretization mistakes in the vertical, so it doesn't conflict with your argument. But this explanation does explain why modelers have focused on improving dx rather than dz even though the latter is more cost effective, as you point out. It may also be worth mentioning the theoretical discussion about the need to change both vertical and horizontal resolution at the same time from [Lindtzen and Fox-Rabinowitz (1989; MWR)](). It is funny how nobody actually links dx and dz when changing resolution even though we know we should.

9. A minor point, but your argument that we should choose resolution based on what we can afford rather than some theoretical panacea (L325) only works if you've formulated your parameterizations in a way that works across all resolutions. Jumping from 100 km to 3 km dx was largely motivated by the sentiment that gray-zone convection must be avoided at all costs.

10. L338: Using emergent constraints in your cost function is a great idea *if you're positive they are real constraints.*

11. P. 16: it strikes me that your "ML as an inverse problem" is very similar to climate model "autotuning", which is being pursued by a lot of groups right now. It may be worth comparing and constrasting your approach against autotuning.

12. I felt like you were glossing over the difficulty of ML as an inverse problem when you have several uncertain parameterizations you are trying to optimize but only have net atmospheric state as your input and net state change as your tuning target. At best there are probably several optimal solutions and at worst your training data is insufficient to predict appropriate behavior. A bit of discussion about why you think this problem is tractable would be appreciated.

13. L421: I really like the idea that we should use ML on detailed subprocesses rather than entire large chunks of the model. I think about this often for microphysics: we have a good sense of what controls each of these detailed processes and we can see that the whole spectrum of conditions these subprocesses will face are probably already being experienced in current climate, so I have a lot more confidence using ML on them to predict future climate. I think you could go a bit further on this point by saying that the choice of which parts of the model will be replaced by ML needs to be made using expert judgement that the process of interest will be climate-invariant and sufficiently sampled in the current climate.

14. I think observational uncertainty is a critical aspect of model optimization, but you don't mention it.

15. L481: I'm a big fan of the idea that climate models need to get a lot better because they must be used for decision support. I like your framing for how we level up these codes. I

would add that exhaustive unit testing and convergence tests are also needed to provide needed confidence in our predictions.

Proofreading:
1. L56: missing "t" at the end of "Turing test"
2. L102: "balance of TOA radiative energy fluxes must also be closed" seems like an obtuse way of saying that energy must be conserved.
3. Fig 1 caption "averaged over the same simulation length" – aren't the AMIP cases actually averaged over the same *dates*? You're not comparing AMIP from 1979 against ICON from 2020, are you?
4. Also for Fig 1, I found "LL, MM, HH" to be confusing terminology. You could just as easily have used titles providing the actual dx for each run. It is also unclear whether you're comparing the ensemble-average AMIP result against a single run from IFS or ICON in those right-hand graphics. I'm surprised that km-scale models have worse precip RMSE than coarse models. This isn't what I've seen… which makes me wonder whether you're comparing ensemble-mean skill (which tends to be superior to individual models) against a particular instance.
5. L200: typo – identifying should be identified.
6. Wherever you talk about the loss function, you add "(1)" afterwards. I found this distracting because I kept thinking you were going to start an enumerated list. I think all readers will know what you're talking about just by mentioning the loss function without referring to the equation.
7. L240: Ad citations for pioneering EDMF papers rather than just your group's recent work here?
8. L337: "This" is an unclear antecedent
9. L370: including an example where the forecast loss function is optimized but climate isn't would be useful. It is easy to think that if you do a good job in each timestep, you will do a good job overall because climate is just the collection of timestep-level results. The obvious counter-example is if you are biased a tiny bit in the same direction every timestep.
10.

---

## Author Comment (AC1)

**Reviewer #2**

This paper presents an opinion on how climate models can be significantly improved by merging traditional model development (improved parametrizations and higher resolution) with AI techniques. I think it is broadly in line with what many operational centres are already trying to do, i.e. augment models with AI instead of replacing them entirely. It is nonetheless valuable to document this in the literature for those who aren't already involved, and therefore I think the paper is an important contribution. It's also well written and I enjoyed reading it.

Thank you for the positive assessment and helpful comments, which led to a number of clarifications in the revision. (However, we are not aware of any operational center that is actually pursuing the approach we advocate, to learn about parameterizations, including functions, from data.)

I have several comments below, which I think the author should consider in producing a revised manuscript.

L56 - typo - "climate Turing test"

Fixed.

L85 - suggest re-wording this slightly - even at kilometer scale, most parametrizations (radiation, cloud macrophysics, cloud microphysics, turbulence, shallow convection, orographic drag) are still required - in fact it's only really deep convection which can plausibly be removed at this scale!

We have amended the sentence in question to read, "More recently, there have been calls to prioritize resolution increase, aiming to achieve kilometer-scale resolutions in the horizontal, with the expectation that this would alleviate the need for subgrid-scale process parameterizations, such as those for deep convection, and substantially increase the reliability of climate predictions."

L105-108 - similar to TOA, the surface precipitation is also required to be accurate to close the water budget, which is critically important for the long-term behaviour of climate models, especially in fully-coupled Earth-systems. Might be worth mentioning this.

We now mention that "precipitation rates are of significant importance as they are part of what closes the water balance" (l. 115). (However, while a closed water balance is helpful, it is less important than a closed energy balance, given that oceans can serve as a large water

reservoir. Coupled climate models that do not conserve water have been successfully used for decades.)

L195 - this is quite a controversial statement that I'm not sure many people would agree with (I certainly don't) - progress undoubtedly is being made, cloud and convection schemes now are measurably better than 10 or 20 years ago when the cited papers were published. I would suggest rephrasing this - what I think is true to say is that progress is not as quick as we would like. The following paragraphs discuss the method by which most groups are making progress in this area (and have been for many years), so it feels slightly disingenuous to present these as new solutions to the problem, when I think they have been known as the solutions by parametrization developers for many years. The issue is that doing as suggested is hard, hence takes a long time.

Our statement is just that "the process-based approach to modeling convection and clouds is *widely perceived* as being deadlocked" (emphasis added, and we have added a more recent reference in the revision). This still seems true to us, even though, of course, we are subsequently making clear that we should reboot, not give up on, the process-based approach.

L252-256 - whilst I agree with the point being made here about developing scale-aware parametrizations, I'm not really sure the model results presented in Figures 1 & 2 are really a compelling argument for it. All major NWP centres run kilometer scale models without convective parametrizations and see huge improvements in NWP skill from doing so. Super-parametrized climate models have similarly shown measurable improvements in model skill relative to traditional parametrizations. Therefore to state on the basis of one bad model that this "approach has not achieved the anticipated success" seems like cherry picking to support the argument, when actually the weight of evidence shows that turning off the deep convective parametrization is better than including it, although could undoubtedly be improved further by scale-aware approaches. I suggest rephrasing this.

We agree that higher resolution has led to improvements in NWP. However, this does not necessarily translate into improvements in climate simulations, which are our focus. Figure 1 summarizes the current state of the art with respect to resolution benefits in climate simulations by showing the only two km-scale simulations run for at least 4 years (that we know of) to publicly release their output (in addition to two single-model resolution hierarchies from HighResMIP). This is an update compared to the previous version, that showed the previous cycle of nextGEMS, with 1 and 2 years of data instead of 4 and 5. The

new version of the figure paints these models in a slightly more positive light, but still does not show large improvements compared to the lower-resolution models.

In section 2, in the context of the discussion of Figure 1 to which the statement in question here refers, we have added: "In numerical weather prediction, enhanced horizontal resolution has led to improvements, for example, in rainfall predictions on timescales from hours to days (Clark et al., 2016). However, whereas assimilation of data at the initialization of a forecast continuously pulls numerical weather predictions close to the climate attractor, long-term climate simulations require a realistically closed energy balance to remain on the climate attractor. This balance also depends on dynamics at scales well below 1 km (e.g., in tropical low clouds, which are crucial for climate but less important for weather prediction), making it less clear that increased resolution by itself results in better climate simulations."

Sect 4 - whilst I agree with what is being said here, it also neglects that there are important aspects of the climate which are not driven by small scale turbulence, e.g. land-sea contrasts, orography, land-surface type, SST pattern. These aspects become better resolved at higher resolution, and in turn leading to improvements in climate model skill which are not simply governed by the energy spectra. This would be worth mentioning.

In the final paragraph of section 4 summarizing the resolution discussion, we added: "While increasing resolution helps by gradually improving the resolution of turbulent dynamics and better resolving surface topography, gravity waves, and land-sea contrasts, the 1000-fold increase in the computational cost in going from O(10 km) to O(1 km) is unlikely to justify the benefits (Wedi et al., 2020)."

L337-341 - this statement is the one that worries me most in the paper. The whole point of emergent constraints is that they are emergent, i.e. they appear in climate models not because they have been programmed to be there, but because they arise as a function of the underlying model physics leading to their emergence in the same way as it does in reality. As soon as we start to pre-program emergent constraints into the model, they lose all meaning and usefulness. It may give the model a better skill when compared to past observations, but this is no guarantee of future success, since we cannot know how the emergent constraint will evolve in a changing climate.

If higher-order statistics such as covariances between SST and cloud cover are informative about a model's response to climate change, our view is that they should be used in model calibration, to obtain the best possible model and to quantify its uncertainties, in much the same way that seasonal temperature changes (and other first-order statistics) are generally used. We added footnote 3, stating, "If emergent constraint statistics are used during loss minimization, they can no longer serve as retrospective constraints on the response of the model to perturbations…"

---

## Author Comment (AC2)

**Reviewer #1**

We thank the reviewer for the thoughtful comments, which led to a number of clarifications in the paper.

1.

> I believe that you shouldn't trust a climate model prediction that you don't understand conceptually. This is particularly necessary for climate modeling because we can't validate any of our predictions until it is too late. Accepting anything on blind faith from a black box model seems like a recipe for disaster. This requires a convergence between episteme and techne which is different from your framing around line 25. Regarding line 248, Bjorn has told me that not using deep convection is strongly motivated by a desire to understand what his model is doing rather than just because it makes the simulation better (which I think most km-scale modelers at this point believe is not necessarily true)

> We generally agree, and we added a statement about the importance of interpretability for trust in climate models and predictions in section 6.

> However, what conceptual understanding means is not always clear cut, and the line between understanding and non-understanding does not necessarily coincide with that between process-based and data-driven models. For example, many microphysics parameterizations have plausible conceptual story lines. But parameters and implementation details such as limiters are not always well constrained, with subtle changes in parameters or implementation details sometimes having large and poorly understood effects on the climate response of a model (e.g., Zhu et al. 2022).

2.

> I felt that the assessment of km-scale models on p. 7 (and, to a lesser extent, high-res models on p.6) was a bit unfair. My feeling is that conventional GCMs have been optimized and tuned for decades but these higher-resolution analogues are still new and generally haven't been well tuned. I think they have a lot of room for improvement. It is hard to say at this point how much benefit they will provide, but they will certainly be better than the versions you're analyzing. One particular challenge is that they are too expensive to tune, which both supports my claim that they have a lot of potential for

improvement but also your point that km-scale models may be too expensive to be practical.

We agree that calibration of small-scale process parameterizations is essential at any resolution, including at km-scales.

We emphasize the need for appropriate parameterization design and calibration in l. 160. Calibration, by hand or with data assimilation/ML tools, requires large ensembles of climate simulations. At the end of section 4, we now emphasize more strongly that generating these large ensembles is currently not feasible at 1 km resolution, but it is feasible at resolutions in the 10-50 km range (especially with climate models that run on accelerator platforms).

3.

Paragraph between line 190 and 195: my personal feeling is even stronger than your argument here – I don't think it's clear that it will ever be possible to adequately parameterize clouds from variables available on the grid scale. Necessary information may simply not be available. I can't think of how to edit your text to express this, so just adding as a comment.

We do not assume that the subgrid-scale (SGS) information can necessarily be inferred from grid-scale information (as in traditional *diagnostic* process-based parameterizations, or most ML-based parameterizations that have been proposed). Instead, parameterizations can consist of sets of auxiliary prognostic equations that carry information about SGS quantities, including memory for them. We are now making this clearer in the revised paragraph starting in l. 277.

4.

L221: When you say "conditional averaging", I think you mean that you will break terms in the governing equations into summands satisfying one condition or another. Just averaging over one particular condition (e.g. only for updrafts) generally does not result in a statement equivalent to the original governing equations. I've had postdocs go astray this way.

We revised the statement by adding, "for example, resulting in distinct equation sets for coherent structures like updrafts and their more isotropically turbulent environment."

5.

Numerated item beginning on L220: I would add that carefully applying scaling arguments to make simplifications and being explicit about the simplifications you're making is critical for readers to understand what you're doing and to be able to assess how much trust they should have in what you do. It may turn out that some assumption you make (like the PDF for subgrid variability) turns out to be inappropriate in some edge case and having those assumptions be clearly listed will help in tracking down these issues.

We added, "Whatever approach is adopted, it is crucial that assumptions are explicitly laid out and subject to empirical validation and revision."

6.

To amplify the last comment, I believe that assumed PDF shape and in particular assumed covariances between variables will be central to the skill of the kind of model you're advocating.

Agreed. (In the EDMF scheme we mention, we model covariances explicitly, with separate equations.)

7.

Discussing surrounding the list of desired properties for parameterizations starting around L219: I think you're missing the possibility for covariance between variables within processes and particularly sub-grid scale interactions between processes. For example, condensation is nonlinearly stronger in portions of a grid cell with stronger updrafts, which wouldn't be captured in models where condensation is performed in microphysics rather than turbulence schemes. I also like Devine et al (2006; GRL), which points out that interactions between convective transport and sub-grid scale spatial variations in DMS are critical for getting cloud microphysics right. All these covariances between processes are things that improved resolution fixes, but would be hard to parameterize without having a single really complicated parameterization that does everything.

We added a fourth point, with some detail on the importance of coupling parameterization schemes for different processes consistently.

8.

Paragraph starting on L304: It is interesting that most modeling centers have found that decreasing dx provides better simulation skill but decreasing dz generally makes the model worse (at least without a ton of extra work). I think this is because the model is actually *more* sensitive to vertical grid and gets its skill from tuning rather than accurate discretized equations, and because it is easy to make discretization mistakes in the vertical, so it doesn't conflict with your argument. But this explanation does explain why modelers have focused on improving dx rather than dz even though the latter is more cost effective, as you point out. It may also be worth mentioning the theoretical discussion about the need to change both vertical and horizontal resolution at the same time from Lindtzen and Fox-Rabinowitz (1989; MWR). It is funny how nobody actually links dx and dz when changing resolution even though we know we should.

We added the Lindzen and Fox-Rabinowitz reference and added that " $\Delta z$ must also be considered and typically should scale with $\Delta x$ \citep{Lindzen89a}; however, existing process-based parameterizations are often manually calibrated to a specific vertical resolution, resulting in a reluctance to increase vertical resolution alongside horizontal resolution in practice."

9.

A minor point, but your argument that we should choose resolution based on what we can afford rather than some theoretical panacea (L325) only works if you've formulated your parameterizations in a way that works across all resolutions. Jumping from 100 km to 3 km dx was largely motivated by the sentiment that gray-zone convection must be avoided at all costs.

We agree that parameterizations ideally should be scale aware, and at the minimum should be appropriate for a given model resolution. Diagnostic parameterizations for convection, without SGS memory, are inappropriate for the convective gray zone, where scale separation disappears. However, this does not mean that parameterizations that are adequate for the convective gray zone cannot be designed. In fact, given the need to parameterize smaller than km-scales, they appear necessary.

We now state explicitly (in l. 370) that "It will remain crucial to make parameterizations as resolution-independent (``scale aware'') as possible."

10.

    L338: Using emergent constraints in your cost function is a great idea *if you're posi-ve they are real constraints.*

    It seems the reviewer is referring to the risk of spurious correlations between an emergent constraint statistic and the climate response of a model (e.g., ECS). This is indeed a serious concern when emergent constraints are used retrospectively.

    We added footnote 3, stating, "In retrospective studies, there is a risk in using emergent constraints because the correlation between emergent constraint statistics and the climate response may be spurious (Caldwell et al. 2014, 2018). When using emergent constraint statistics during loss minimization, by contrast, the statistics at worst may merely be uninformative about model parameters and processes."

11.

    P. 16: it strikes me that your "ML as an inverse problem" is very similar to climate model "autotuning", which is being pursued by a lot of groups right now. It may be worth comparing and constrasting your approach against autotuning.

    "Autotuning" is used in various ways, typically referring to the optimization of parameters affecting computational performance of a model. It seems the reviewer here is referring to various approaches to calibrating scalar parameters in climate model parameterizations. We added (l. 494): "It moves beyond automatically calibrating scalar parameters in climate models (Zhang et al., 2015; Couvreux et al., 2021; Hourdin et al., 2023) to encompass higher-dimensional parameter spaces, including those relevant to deep learning approaches." (That is, we want to learn not just scalar parameters, but also parametric or even non-parametric functions from data.)

12.

    I felt like you were glossing over the difficulty of ML as an inverse problem when you have several uncertain parameterizations you are trying to optimize but only have net atmospheric state as your input and net state change as your tuning target. At best there are probably several optimal solutions and at worst your training data is insufficient to predict appropriate behavior. A bit of discussion about why you think this problem is tractable would be appreciated.

    Yes, this is a valid and serious concern. We added: "As in many inverse problems, minimizing the loss function is often an ill-posed problem with many possible solutions,

which may be sensitive to small changes in the data (Tarantola, 1987; Hansen, 1998; Iglesias et al., 2013). This requires regularization, for example, through the use of prior information on the parameters ∨ to select "good" parameter sets among the many that may minimize the loss. Such prior information may be obtained, for example, by pre-training on computationally generated data, which can be more detailed than observational data (Lopez-Gomez et al., 2022)."

13.

L421: I really like the idea that we should use ML on detailed subprocesses rather than entire large chunks of the model. I think about this often for microphysics: we have a good sense of what controls each of these detailed processes and we can see that the whole spectrum of conditions these subprocesses will face are probably already being experienced in current climate, so I have a lot more confidence using ML on them to predict future climate. I think you could go a bit further on this point by saying that the choice of which parts of the model will be replaced by ML needs to be made using expert judgement that the process of interest will be climate-invariant and sufficiently sampled in the current climate.

Agreed. We expanded on this point in a revision of the final paragraph of section 5.

14.

I think observational uncertainty is a critical aspect of model optimization, but you don't mention it.

Yes, observational uncertainty is important. We had briefly mentioned it in the text following equation (1), in the discussion of the covariance matrix in the loss function. We now re-emphasize this in l. 412-414.

15.

L481: I'm a big fan of the idea that climate models need to get a lot better because they must be used for decision support. I like your framing for how we level up these codes. I would add that exhaustive unit testing and convergence tests are also needed to provide needed confidence in our predictions.

Agreed. We added a new paragraph to this effect toward the end of section 6 (l. 583).

Proofreading:

1. L56: missing "t" at the end of "Turing test"

Fixed.

2. L102: "balance of TOA radiative energy fluxes must also be closed" seems like an obtuse way of saying that energy must be conserved.

Added a statement on energy conservation.

3. Fig 1 caption "averaged over the same simulation length" – aren't the AMIP cases actually averaged over the same *dates*? You're not comparing AMIP from 1979 against ICON from 2020, are you?

AMIP is averaged over the last 4 or 5 years of the simulation (to compare with IFS and ICON, respectively), i.e. 2011-2014 and 2010-2014. IFS and ICON cycle 3 simulations are for 2020-2024, when AMIP is not available. Since we are looking at the skill for reproducing the observed climatology over 2001-2020, our analysis is not sensitive to which years are used (we checked for other choices). We were sampling multiple choices of years to average over in the previous version when nextGEMS cycle 2 had only 1 and 2 years of data for each model, but we found this no longer influenced the answer once extending to the longer averaging period.

4. Also for Fig 1, I found "LL, MM, HH" to be confusing terminology. You could just as easily have used titles providing the actual dx for each run.

Amended as suggested.

It is also unclear whether you're comparing the ensemble-average AMIP result against a single run from IFS or ICON in those right-hand graphics. I'm surprised that km-scale models have worse precip RMSE than coarse models. This isn't what I've seen... which makes me wonder whether you're comparing ensemble-mean skill (which tends to be superior to individual models) against a particular instance.

The single-runs from IFS and ICON are being compared against the median over AMIP of the same statistics computed in individual AMIP runs. To make this clearer, we updated one sentence of the caption to read "CMIP and AMIP rms errors represent median values of the RMSE computed separately in each of the included models."

5. L200: typo – identifying should be identified.

"Identifying" is correct.

6. Wherever you talk about the loss function, you add "(1)" afterwards. I found this distracting because I kept thinking you were going to start an enumerated list. I think all readers will know what you're talking about just by mentioning the loss function without referring to the equation.

We prefer to keep the equation number for clarity and specificity in some (but not all) places where the loss function is mentioned.

7. L240: Ad citations for pioneering EDMF papers rather than just your group's recent work here?

We added references to some of the pioneering EDMF papers in the first point on p. 10.

8. L337: "This" is an unclear antecedent

Clarified as "this covariance."

9. L370: including an example where the forecast loss function is optimized but climate isn't would be useful. It is easy to think that if you do a good job in each timestep, you will do a good job overall because climate is just the collection of timestep-level results. The obvious counter-example is if you are biased a tiny bit in the same direction every timestep.

Added a sentence with a reference to Schreiber et al. (2013).